# Geminga SNR: Possible Candidate of Local Cosmic-Ray Factory (II)

**Bing Zhao [1], Yiqing Guo [2,3],* and Xunxiu Zhou [1],***

[1]   School of Physical Science and Technology, Southwest Jiaotong University, Chengdu 610031, China
[2]   Key Laboratory of Particle Astrophysics, Institute of High Energy Physics, Chinese Academy of Sciences, Beijing 100049, China
[3]   University of Chinese Academy of Sciences, Beijing 100049, China
*   Correspondence: guoyq@ihep.ac.cn (Y.G.); zhouxx@home.swjtu.edu.cn (X.Z.)

**Abstract:** Accurate measurements of the energy spectrum and anisotropy can help us discover local cosmic-ray accelerators. Our recent works have shown that spectral hardening above 200 GeV in the energy spectra and transition of large-scale anisotropy at ∼100 TeV are of an unifying origin. Less than 100 TeV, both spectral hardening and anisotropy explicitly indicate the dominant contribution from nearby sources. Recent observations of CR anisotropy suggest that this phase is consistent with the locally regular magnetic field (LRMF) of the interstellar boundary explorer (IBEX) below 100 TeV. In this work, we further investigate the parameter space of sources allowed by the observational energy spectra and amplitude and phase of dipole anisotropy. To obtain the best-fit source parameters, a numerical algorithm is to compute the parameter posterior distributions based on Bayesian inference. We found that by combining the observations of the energy spectrum and anisotropy, the parameters of the model can be well constrained. The LRMF and the effect of the corresponding anisotropic diffusion are considered in this work. Finally, the phase results' right ascension $(R.A.) = 3.2$ h below 100 TeV was obtained by fitting, which is in general agreement with the experimental observations. Since the Geminga SNR is very close to the mean of the fitted parameters, it could be a candidate for a local cosmic-ray accelerator.

**Keywords:** cosmic rays; dipole anisotropy; MCMC





## 1. Introduction

Supernova remnants (SNRs) or pulsars have been proposed as the accelerated sources of galactic cosmic rays (GCR) [1]. The characteristics of the models depend on the properties of the interstellar medium (ISM) through which the particles travel. For example, the galactic magnetic field, both its regular and turbulent components are related to propagation parameters in such models. The single-source model, in which one or a few nearby young sources make a non-negligible contribution to the the solar system spectrum, was originally proposed for the sharpness of the knee region at ∼3–4 PeV in the all-particle spectrum [2]. With increasingly advanced instruments being put into use, measuring accuracy has been promoted greatly, and more novel features in the energy spectrum have been uncovered. The single-source model and its extension, the local source model, are widely used to interpret various observational phenomena. Usually, the propagation of CR from the nearby source is time-dependent, and the propagated spectrum resembles a bump-like structure, which is deemed as an excess of CR flux. Thus, the local pulsar or SNR could be the natural origin of the positron excess above 10 GeV [3–6], the spectral hardening of nuclei above 200 GeV [7–13], and the ensuing softening at ∼20 TeV [14–16]. Meanwhile, it could also account for the break in the all-electron spectrum at TeV energies [17–20].

However, due to the diffusive character of CR propagation in the Galaxy, it is hard to locate their acceleration sites by tracing back the arrival directions of CR, Furthermore, in the traditional propagation model, the predicted anisotropy amplitude from background

SNRs far exceeds the measurements, which is only about $10^{-4}$–$10^{-3}$ [21]. The local source could effectively lower the amplitude, if it lies close to the direction of the anti-Galatic center [22,23].

In recent works, we established a unified model to explain both observed spectral features and anisotropy [24,25]. We find that the amplitude transition and phase flipping in the dipole anisotropy map have a common origin with the spectral hardening of nuclei above 200 GeV and ensuing falloff at $\sim$20 TeV. Less than 100 TeV, the anisotropy and spectral features are dominated by the local source. The position of the local source is close to the direction of the anti-Galatic center and far from the Galactic disk. We find that the Geminga SNR at its pulsar's birthplace could be a prime candidate, but considering the phase of dipole anisotropy, the Geminga SNR position seems to be inconsistent with the observations at lower energy. Recent observations of CR anisotropy suggest that this phase is consistent with the locally regular magnetic field (LRMF) of the interstellar boundary explorer (IBEX) below 100 TeV [26]. The emission of energy neutral atoms is enhanced along a circular ribbon that defines a magnetic field, along with galactic longitude $l \simeq 210.5°$ and galactic altitude $b \simeq -57.1°$ with an uncertainty of $\sim$1.5° [27], which is closer to the observed phase of the low-energy dipole anisotropy .

The diffusion of CR in a magnetic field with a significant ordered field component and the orientation with respect to this field is usually anisotropic [28] and is usually stronger in the field-parallel direction and weaker in the perpendicular direction, and the perpendicular diffusion coefficient $D_\perp$ is scaled to be a fraction of $D_\parallel$. In fact, the Geminga pulsar has long been considered as a local positron source since the discovery of the increasing positron fraction above 10 GeV. Recently, the HAWC experiment measured the extended TeV gamma-ray emission of Geminga and PSR B0656+14 pulsars [29]. The inferred diffusion coefficient near the $\gamma$-ray emission region is far lower than the standard value derived by fitting the B/C ratio. It also suggested the diffusion coefficient is energy-dependent.

So, in this work, we aim at the parameter space of cosmic-ray sources permitted by the observed energy spectra and anisotropy. To perform the elaborate scan of the parameter space of sources, the multinest package [30,31], based on Bayesian inference, is applied. By fitting the energy spectra and anisotropy amplitude, the permissible space of location and the age of the local source is greatly reduced. Our study further demonstrates that the Geminga SNR could be the best candidate for a local cosmic-ray source.

The rest paper is organized as follows: in Section 2, the propagation model is briefly introduced. Section 3 presents the simulation results, and Section 4 is reserved for the conclusion.

## 2. Model Description

### 2.1. Propagation Model

The spatial-dependent propagation (SDP) model of cosmic rays has been increasingly adopted in recent years. A two-halo model (THM) was introduced [32] to explain the spectral hardening of both proton and helium above 200 GeV [10], and was also applied to secondary and heavier components [33–37], positrons and electrons [38], diffuse gamma-ray distribution [39], and large-scale anisotropy [24,25]. For a comprehensive introduction, one can refer to [35,36].

The whole diffusive halo is divided into two parts. The inner halo (IH) contains the galactic disk and its surrounding region, in which the diffusion coefficient is determined by the distribution of the CR sources in the Milky Way cantilever. The other part of the diffusive region is named the outer halo, where the diffusion coefficient is regarded as rigidity-dependent. The size of IH is represented by its half thickness $\xi z_h$, whereas the OH region's is $(1 - \xi)z_h$. The diffusion coefficient $D_{xx}$ can be expressed as

$$D_{xx}(r, z, \mathcal{R}) = D_0 F(r, z)\beta^\eta \left( \frac{\mathcal{R}}{\mathcal{R}_0} \right)^{\delta_0 F(r,z)}. \tag{1}$$

where

$$F(r,z) = \begin{cases} g(r,z) + [1 - g(r,z)] \left( \dfrac{z}{\xi z_0} \right)^n, & |z| \le \xi z_0 \\ 1, & |z| > \xi z_0 \end{cases},$$

(2)

with $g(r,z) = N_m / [1 + f(r,z)]$. $f(r,z)$ is the source density distribution, which can be parameterized as

$$f(r,z) = \left( \frac{r}{r_\odot} \right)^\alpha \exp \left[ -\frac{\beta(r - r_\odot)}{r_\odot} \right] \exp \left( -\frac{|z|}{z_s} \right).$$

(3)

with $r_\odot \equiv 8.5$ kpc and $z_s = 0.1$ kpc. $\alpha$ and $\beta$ are taken as 1.69 and 3.33, respectively [40]. The propagation of CR from local point source is time-dependent. As for the instantaneous injection, the spatial distribution is

$$\psi(\mathcal{R}, \vec{r}, t) = \frac{Q(\mathcal{R})}{(4\pi D_{xx} t)^{3/2}} e^{\frac{-(\vec{r} - \vec{r}')^2}{(4 D_{xx} t)}}.$$

(4)

The energy spectra at sources are assumed to have a power-law of rigidity plus an exponential cutoff:

$$Q(\mathcal{R}) \propto \mathcal{R}^{-\nu} \exp \left( -\frac{\mathcal{R}}{\mathcal{R}_c} \right).$$

(5)

*2.2. Anisotropic Diffusion*

We adapted an isotropy diffusion method in our pre-work [41], but with the diffusion of CR in a magnetic field with a significant ordered field component, and the orientation with respect to this field is usually anisotropic [28], i.e., stronger in the field-parallel direction and weaker in the perpendicular direction . So, this can be described by a diffusion tensor that is local, that is, at the field-aligned coordinated system, we obtained a diagonal:

$$\mathbf{D} = \begin{Bmatrix} D_{//} & 0 & 0 \\ 0 & D_\perp & 0 \\ 0 & 0 & D_\perp \end{Bmatrix}.$$

(6)

A new tensor can be obtained that forms in the Cartesian coordinate system by rotating the coordinate system, which can be written as

$$D_{ij} = D_\perp \delta_{ij} + (D_\parallel - D_\perp) \cdot \frac{B_i \cdot B_j}{|B|^2}.$$

(7)

where $i$, $j$ present the different directional components, $D_\parallel$ and $D_\perp$ are the diffusion coefficients aligned parallel and perpendicular to the regular magnetic field, and $B_i$ is the magnetic field strength in this direction. Moreover, the perpendicular diffusion coefficient $D_\perp$ is scaled to be a fraction of $D_\parallel$, i.e., $D_\perp = \epsilon \cdot D_\parallel$, where the $\epsilon$ is assumed to be in the range 0.01 to 0.1 for the galactic proton with GeV energies [28]. As calculated in the model, the large-scale dipole anisotropy is proportional to the spatial gradient of CR density $\nabla \psi$ and the diffusion coefficient tensor $\mathbf{D}$,

$$\delta^* = \frac{3\mathbf{D}}{c\psi} \cdot \nabla \psi$$

(8)

In this work, the diffusion-reacceleration (DR) propagation model is adopted. The observation result of the IBEX shows that the magnetic field exists along with galactic longitude $l \simeq 210.5°$ and galactic altitude $b \simeq -57.1°$ with an uncertainty of $\sim 1.5°$ [27], corrected by the predicted Compton-getting shift [22]. The dipole direction is very consistent with the magnetic field direction mentioned by IBEX. So, the magnetic field whose direction is derived from IBEX observations is used to compute the anisotropy, and the

numerical package DRAGON [42] is used to solve the diffusion equation to obtain the CR distribution.

## 3. Results

As for the spatial-dependent propagation (SDP) model, we use the same parameters $D_0$, $\delta_0$, $N_m$, $\xi$, $n$, $v_A$, and $z_h$ as in the previous work [41], and we list them in Table 1. We study injection parameters as well as local source's age and distance, as well as the mag. The MultiNest package is applied to perform the Bayesian inference of the corresponding parameters to obtain their posterior distributions and correlations between those parameters allowed by the observations ([43]). The background and local source parameter set $\vec{\Theta} = \{A^{\mathrm{P}}, \gamma^{\mathrm{P}}, A^{\mathrm{He}}, \gamma^{\mathrm{He}}, q_0^{\mathrm{P}}, \alpha_{\mathrm{P}}, q_0^{\mathrm{He}}, \alpha_{\mathrm{He}}, \mathcal{R}_c, r, t, gl, gb, \epsilon, \delta\}$. The parameters of $A^{\mathrm{P/He}}, \gamma^{\mathrm{P/He}}$ are the normalization galactic background proton/helium flux at 100 GeV and power index of background proton/helium flux, and $q_0^{\mathrm{P/He}}, \alpha^{\mathrm{P/He}}$ are the injection power of local source for protons/helium nuclei, which is set at the rigidity of 1 GV and the power index. $\mathcal{R}_c$ is the cut-off rigidity of local source CR. The other parameters of the $r$, $t$, $gl$, and $gb$ are set as the local source's distance, age, longitude, and latitude in the Galactic coordinate system. $\epsilon$ is the ratio of $D_\perp$ and $D_\parallel$, and $\delta$ is the power index of the diffusion coefficient at the reference rigidity $\mathcal{R}_0$.

**Table 1.** Fitted spatial-dependent propagation parameters.

| $D_0$ [$\mathbf{cm^2 \cdot s^{-1}}$] | $\delta_0$ | $Nm$ | $\xi$ | $n$ | $v_A$ [$\mathbf{km \cdot s^{-1}}$] | $z_h$ [$kpc$] |
|---|---|---|---|---|---|---|
| 4.66 | 0.54 | 0.62 | 0.1 | 4 | 6 | 5 |

The fit data include proton and helium spectra, and the lag-scale dipole anisotropy amplitude. Unlike the pre-work, we are more interested in fitting the phase of the cosmic-ray anisotropy in the magnetic field, we re-consider the dipole anisotropy data of HAWC ([44]) experiment for fitting.

The two-dimensional correlation distributions of the parameters are illustrated in the triangular plot of Figure 1, and marginalized posterior PDFs shown in the diagonal regions. The dark, intermediate, and light-blue lines correspond to $1-\sigma$, $2-\sigma$, and $3-\sigma$ contours, respectively. The posterior distributions of each parameter are listed in Table 2. For the background parameters, $A^{\mathrm{P}}$ and $\gamma_{\mathrm{P}}$ (or $A^{\mathrm{He}}$, $\gamma_{\mathrm{He}}$) are distinctly anti-correlated. This can be understood that since the injection spectrum is softer, the calculated flux at normalization energy 100 GeV is lower and a larger normalization flux is needed in order to fit the spectrum. This the same for $q_0^{\mathrm{P}}$ and $\alpha_{\mathrm{P}}$ (or $q_0^{\mathrm{He}}$ and $\alpha_{\mathrm{He}}$) of the local source.

As can be noticed, the age and distance of the local source have a strong positive correlation. The distant local source needs to be old due to the longer propagation distance performance seeking control . Otherwise, the CR will not propagate to the solar system if the source is too young . Correspondingly, its injection power has to be enhanced when its distance is far away. Therefore, $q_0^{\mathrm{P}}$ and $q_0^{\mathrm{He}}$ are positively correlated with the source's distance $r$. We also found that to explain the proton and helium spectra, the injection power index of the local source is slightly harder than the background. For example, the power index of local protons $\alpha$ is between $-2.2$ and $-1.9$, whereas it is between $-2.40$ and $-2.30$ for the background. This is the same for helium. This has been noticed in our previous works [24]. When the source is young, the shock is very strong; the standard diffusive shock acceleration predicts that the power index is close to $-2$. As the sources become older, the shock becomes weaker and the accelerated spectra of CR become steeper. The background CR are the sum of the contribution of both young and old sources in the Galaxy, so the injection spectra of the background are expected to be steeper than the local one. Furthermore, to fit both energy spectra and anisotropy amplitude, the constraint of the local source's cutoff rigidity is very tight, between 20 and 28 TeV. It seems to have no significant

correlations with other parameters and even does not change with the local source's age and distance.

**Table 2.** Fitted injection parameters of the background and local sources.

| Parameter | | 68% Limits |
|---|---|---|
| Background | $\gamma^P$ | $-2.374 \pm 0.017$ |
| | $A^P$ † | $0.0332^{+0.0022}_{-0.0026}$ |
| | $\gamma^{He}$ | $-2.332^{+0.016}_{-0.014}$ |
| | $A^{He}$ | $0.0385^{+0.0014}_{-0.0017}$ |
| Local source | $\alpha^P$ | $1.978^{+0.037}_{-0.049}$ |
| | $\log_{10}(q_0^P)$ | $52.30 \pm 0.25$ |
| | $\alpha^{He}$ | $1.947^{+0.022}_{-0.038}$ |
| | $\log_{10}(q_0^{He})$ | $52.00 \pm 0.20$ |
| | $R_c$ [TV] | $28.3^{+0.16}_{-0.064}$ |
| | $r$ [kpc] | $0.366^{+0.090}_{-0.063}$ |
| | $t$ [kyr] | $390^{+70}_{-60}$ |
| | gl[°] | $207 \pm 30$ |
| | gb [°] | $-65^{+12}_{-10}$ |
| Anisotropy | $\epsilon$ | $0.26^{+0.16}_{-0.14}$ |
| | $\delta$ | $0.175^{+0.049}_{-0.097}$ |

† unit of normalization $A^{P/He}$ and $q_0^{P/He}$ are GeV$^{-1}$ m$^{-1}$ s$^{-1}$ sr$^{-1}$ and GeV$^{-1}$ s$^{-1}$ respectively.

Figure 2 shows the calculated proton and helium spectra with 1-$\sigma$ errs of fitted parameters, and the lower panel shows the residuals of each data point. The cyan solid line in the figure indicates the background energy spectrum of the galactic background, while the red solid line shows the local source injection spectrum. The black line is obtained by merging the background and local source spectrum. The figure indicated that we can reproduce the proton and helium spectra from experimental observations very well.

Additionally, Figure 3 shows the fitted results of the amplitude and phase of dipole anisotropy. It can be seen that the anisotropy amplitude increases gradually from low to high energy and is mainly dominated by the local source, due to the higher gradient of the cosmic-ray flow intensity in the direction of the local source. However, up ~20 TeV, the amplitude starts to decrease due to the anti-galactic direction of the local-source, and the anisotropy amplitude is gradually dominated by the cosmic-ray intensity in the Galactic center up to the higher energy. The transition from local-source dominated to background-dominated in the amplitude map is a little over 100 TeV. Even if the ARGO-2018 and AS$\gamma$ data are not used, they still fit the observation well at ~100 TeV.

We include in the phase fit of the anisotropy the LRMF direction measured by the IBEX experiment, i.e., 3.23 h, which makes the calculated phase consistent with observations of less than 100 TeV. However, above 100 TeV, the phase suddenly flips from the direction of the local source to the direction of the galactic center, Which is consistent with the change in amplitude.

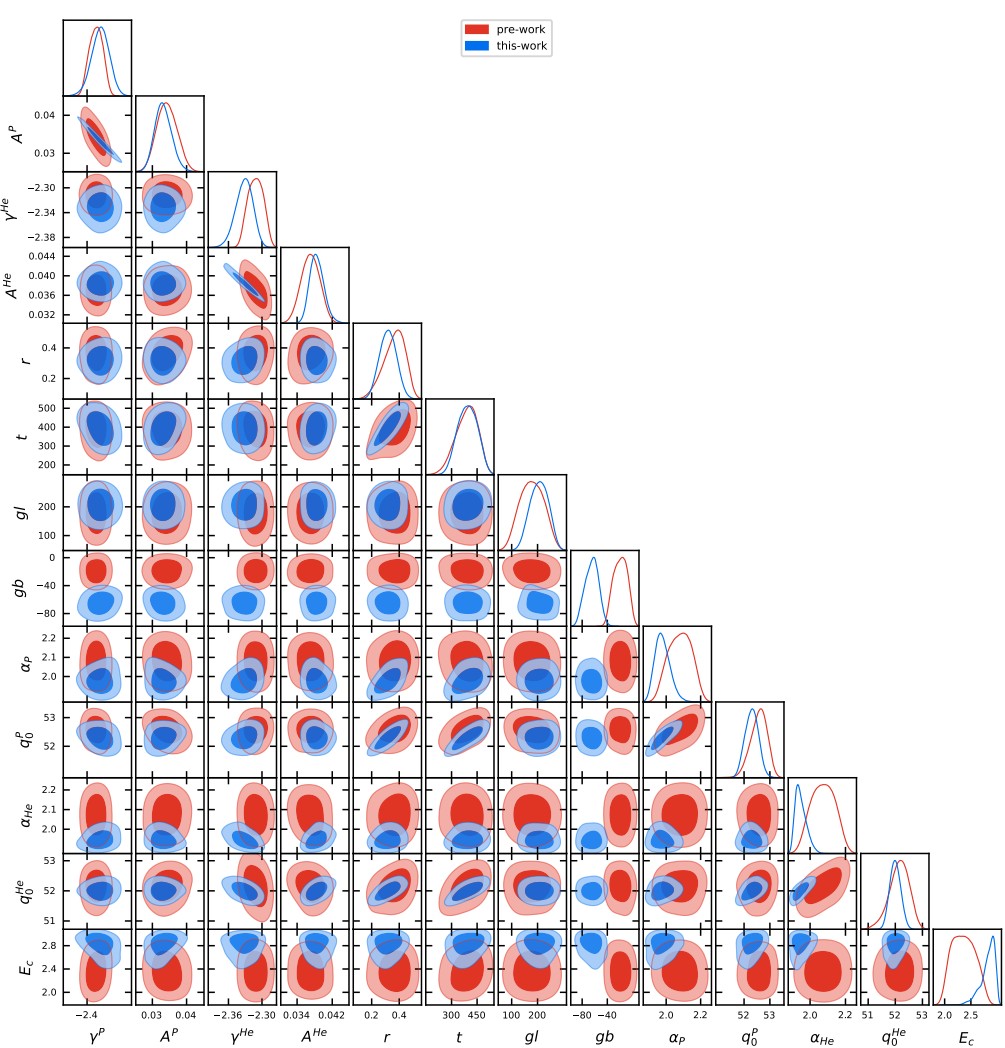

**Figure 1.** 2-dimensional correlation distributions of injection parameters as well as local source's age and distance.

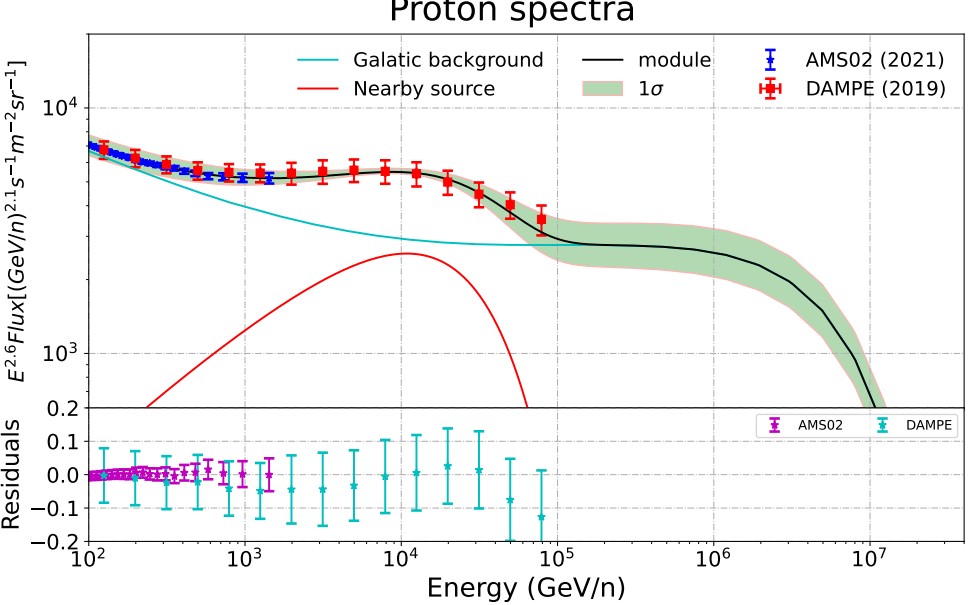

**Figure 2.** *Cont.*

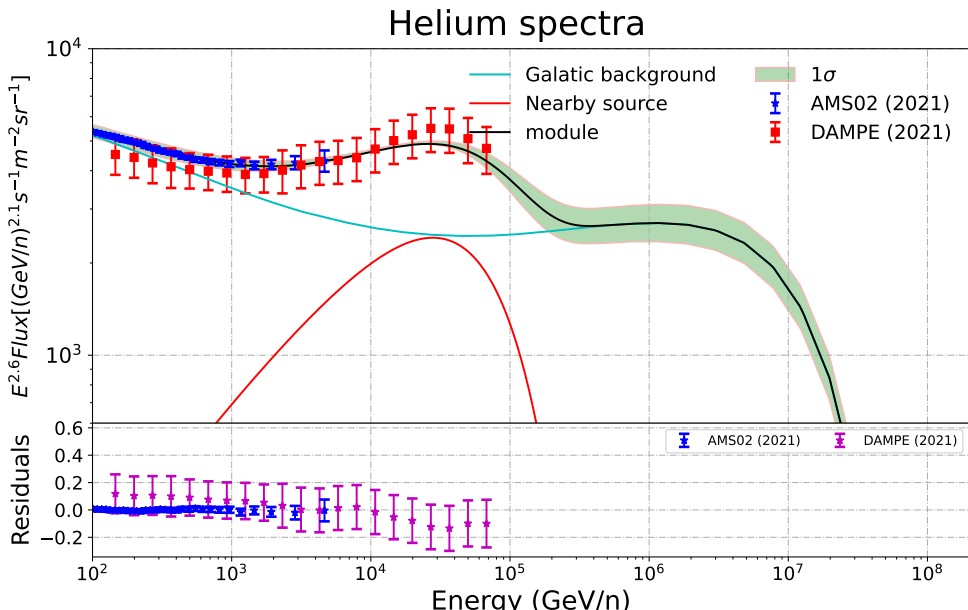

**Figure 2.** Proton and helium spectra fit result by mcmc combined with data. The cyan line is the energy spectrum of the galactic background. The red one is the local source inject spectra. The data points are taken from AMS-02 ([45]) and DAMPE ([16,46]).

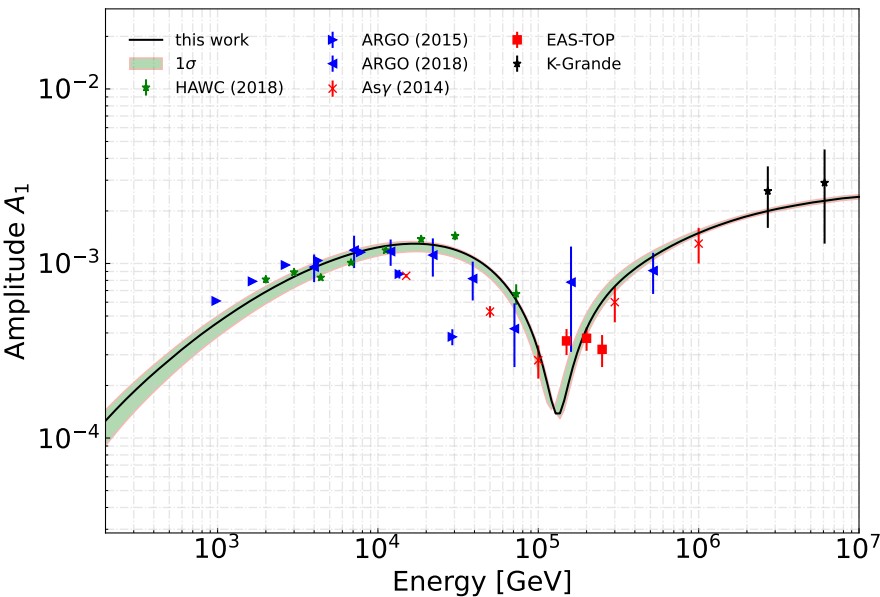

**Figure 3.** *Cont*.

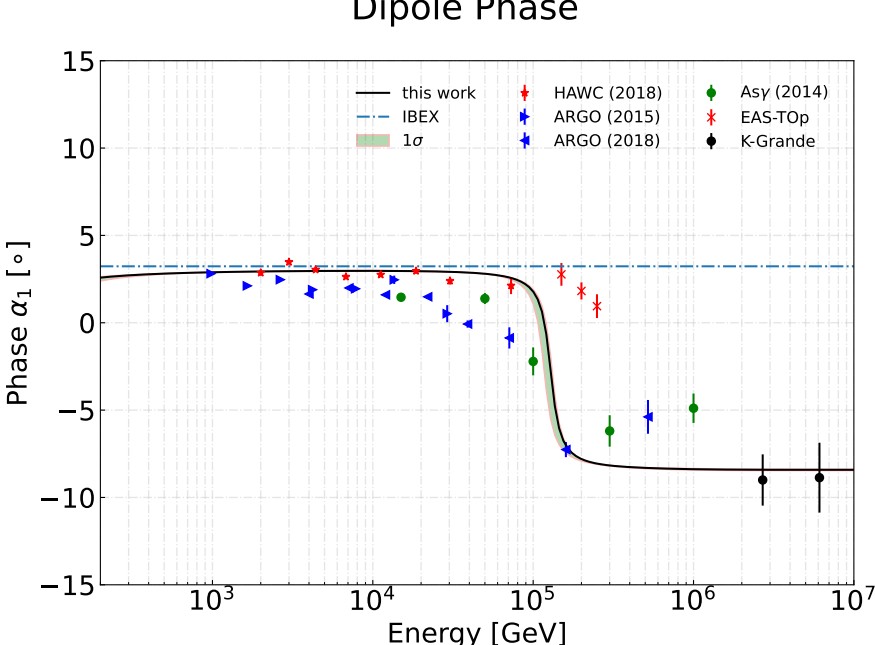

**Figure 3.** Amplitude and phase of dipole anisotropy fit result with best-fit parameters. The data points are taken from HAWC ([44]), ARGO-YBJ ([47,48]), and Tibet ([49]).

When the local source's position and age are constrained, we further attempt to find out the best candidate in the local catalog [43]. Figure 4 shows the 2-D contours of the age and distance of local source, with local SNRs and puslars shown as green circle and blue square points. The Geminga pulsar, J1741-2054, is in the 3-$\sigma$ range, as shown in red and blue squares. Moreover, we also include Geminga SNR, which is taken as the local source to explain both the energy spectra and anisoropy in our previous works. The violet star is the best fit values of age and distance, and the solid lines are the 1-$\sigma$, 2-$\sigma$ and 3-$\sigma$ contours, respectively. As can be seen from the figure, most of the local sources are far from the contours and are thus excluded. Only Geminga-pulsar/SNR and J1741-2054 are within the 3-$\sigma$ regions. The Geminga-pulsar is just at the 3-$\sigma$ contour and the Geminga-SNR is closest to the best-fit value.

Figure 5 shows the top view of the galactic disk, the 2-D contour of the source distance, and the galactic longitude. The red spots indicate SNRs, while green indicate pulsars, and the solid lines are 1-$\sigma$, 2-$\sigma$ and 3-$\sigma$ contours, respectively. The black arrow is the direction of the LRMF, and the green arrow is in the direction of the galactic center. Where the purple cross is the mean of fit center, the star is Geminga-SNR. Nevertheless, the constraints are still very tight, and the allowed sources are only Geminga SNR and Geminga pulsar, as well as G203.0+12.0, which is named the Monogem ring. The local sources at the anti-Galactic center direction are very few; most of them are in the direction of the Galactic center. Additionally, Geminga SNR is very close to the parameters as well as the local source's age and distance.

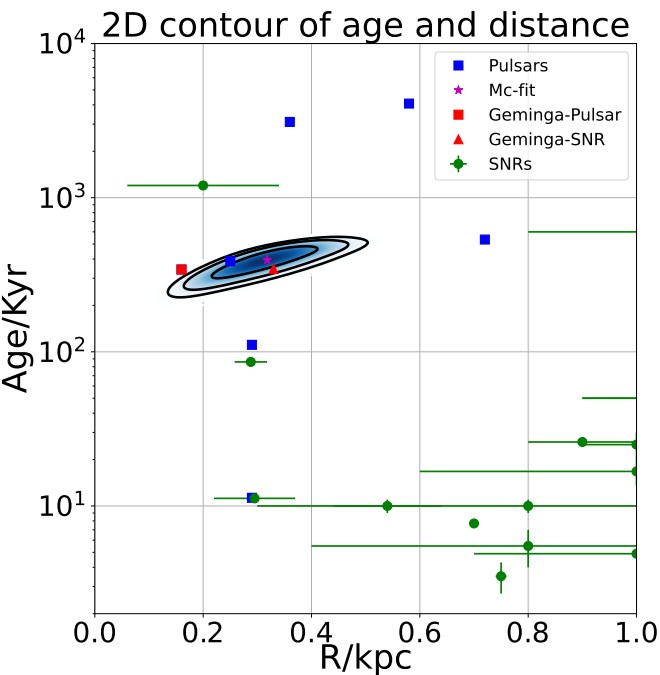

**Figure 4.** 2D contour of age and distance of local source. The violet star is best fit values of galactic longitude and latitude, and the solid lines are 1-$\sigma$, 2-$\sigma$, and 3-$\sigma$ contours, respectively.

## Source Distribuation Map

**Figure 5.** 2D contour of source distribution on galactic disk. The red spots indicate SNRs, green indicates pulsars, and the solid lines are 1-$\sigma$, 2-$\sigma$, and 3-$\sigma$ contours, respectively. The black arrow is the direction of the LRMF, and the green arrow is in the direction of the galactic center. Where the purple cross is the mean of fit center, the star is in the Geminga-SNR location.

## 4. Conclusions

We established a unified scenario, based on SDP + local sources, to interpret the observed energy spectrum and anisotropy of CR nuclei below PeV energy, and to consider the LRMF to enhance the phase fit. We found that for less than 100 TeV, the local source contributes not only to the spectral hardening of 200 GV in the energy spectrum and the

subsequent softening of 20 TeV but also dominates the GCR flow and determines the low-energy amplitude and phase of dipole anisotropy. The parameters of SDP were adapted in this paper, such as $n = 4$, which indicated that the diffusion coefficient in the inner halo was much lower at GeV energies than the diffusion coefficient in the outer halo. In addition, it has weaker energy dependence. All of this results in the significant contribution of the local source at TeV energies. Considering our pre-work, we still infer that the SNR associated with Geminga may be an important local-source candidate.

In this work, we further investigate the injection parameters and local source's position and age in detail using the Bayesian inference tool, MULTINEST. We find that the age and distance of the local source are positively correlated. For a distant local source, its age has to be older. Otherwise, the CR cannot propagate to the solar system if the source is too young. Additionally, the corresponding injection power is enhanced, so $q_0^P$ and $q_0^{He}$ are also expected to be positively correlated with the source's distance $r$. Meanwhile, both energy spectra and anisotropy amplitude severely constrain the local source's cutoff rigidity, which is between 20 and 28 TV.

The energy spectra and anisotropy amplitude also provide strong constraints of local source age and distance to the solar system. Most of local sources have been excluded, and we find that Geminga SNR is very close to the best-fit value. In addition, after considering the LMRF, the phase fit result seems closer to the experimental observation. Since most of sources are in the direction of the galactic center, Geminga SNR is very close to the best-fit value. From these, we infer that the Geminga SNR is the probable candidate of the local source.

**Author Contributions:** Conceptualization, Y.G.; formal analysis, B.Z.; data curation, B.Z.; writing—original draft preparation, B.Z.; funding acquisition, X.Z. All authors have read and agreed to the published version of the manuscript.

**Funding:** This work is supported by the National Key Research and Development Program of China (No. 2018YFA0404202) and the National Natural Science Foundation of China (Nos. 12275279, U2031101).

**Acknowledgments:** Software: DRAGON ([42,50]) available at https://github.com/cosmicrays accessed on 1 February 2023. MULTINEST ([30,31]) available at https://github.com/farhanferoz/MultiNest, accessed on 1 February 2023.

**Conflicts of Interest:** The authors declare no conflict of interest.

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
