# Peer review of "Geminga SNR: Possible Candidate of Local Cosmic-Ray Factory (II)"

_universe, doi:10.3390/universe9020093_

Round 1

Reviewer 1 Report

Attached pdf

Author Response

Thank you for pointing out some manuscript errors and for the suggestions. The responses are listed in the reply letter.

Reviewer 2 Report

The manuscript is an extension of the paper https://iopscience.iop.org/article/10.3847/1538-4357/ac4416 (paper I).  In this work, the anisotropic diffusion is considered in addition to the former model and the main conclusion keep unchanged, the author suggest the undetectd SNR of Geminga pulsar is the most promising candidate for nearby CR accelerators. The paper show potential interests to be published, but before publication the paper need significant revisions to address some key issues. 

Major comments: 

1. In this paper, the anisotropic diffusion with respect to the magnetic field is considered. But it seems the configuration of magnetic field is not well described, I guess the authors assume a uniform magnetic field whose direction is derived from IBEX observations, is this true?  Please describe it in detail in sec.2.2. 

2. When considering the anisotropic diffusion, if the author assume the magnetic field is along the plane, the diffusion perpendicular to the plane would be significantly smaller. But a same D_0 and Z_h are assumed  as in paper I , so I doubt whether the current configuration can fit the B/C ratio, maybe it is better to clarify in the manuscripts. 

3. It is also widely believed the magnetic field in the plane have a coherence length of about 100 pc, which is smaller than the derived distance of Geminga SNR. Does the coherence length play a role in the scenario considered here? I suggest the author make some commets on this issue. 

minor comments: The paper seems not well written and the language require further polishing. Here I list the typos in the abstract:

line 4: 200GeV, should be $200~\rm GeV$, please use Roman for the units and leave a space between number and units, all over the manuscripts. 

line 5:unifying->unified

line 7:  the abbreviation SDP is not defined 

line 8: try->tried

line 10: a numerical packet algorithm -> a numerical algorithm ?

line 12 : We find that by combining the energy spectra and anisotropy data. The sentence is not complete. 

line 13: A phase fit with simulated -> A phase of ? 

line 16: The Geminga SNR -> it 

I suggest the authors to proof read the draft carefully before resubmit. 

Author Response

Thank you for pointing out some errors in the manuscript. The answers to each of the questions are listed in the word file.

Reviewer 3 Report

Referee report on the paper of Bing Zhao et al. 

 "Geminga SNR: Possible candidate of local cosmic-ray factory (II)"

The authors presented a model describing the influence of the nearby cosmic ray source on  the observed spectra and anisotropy of Galactic cosmic rays. Using the spatial-dependent propagation (SDP) model and including anisotropic diffusion they showed that the spectral hardening at 200 GeV, the break at tens TeV and properties of observed anisotropy can be explained in the model of one nearby cosmic ray source. The best candidate is Geminga SNR.

The paper contains new interesting results and can be published after a minor revision. 

Comments:

1) An important assumption is the use of SDP model.  As I understood from eq. (1) the diffusion coefficient in Galactic midplane is 3 times lower at GeV energies than the diffusion coefficient in the outer halo. In addition, it has weaker energy dependence. All this result in large contribution of the local source at TeV energies. It would be good if the authors add several words about it  in section 2.1 or in Conclusion.

2) It is better to give numeric values of the parameters of the model (D_0 and other, those given in the previous paper) (beginning of Sect.3). 

3) It would be nice if the authors give the estimated cosmic ray power of the  nearby source. 

4) There is a misprint in Table 1: "antisopy" instead of "anisot

Author Response

Thank you for pointing out some errors in the manuscript. The answers to each of the questions are listed in the following document.

Round 2

Reviewer 2 Report

Thanks for the authors revision, all my sugegstions have been addressed and I suggest acceptance for publication.